# Physicochemical Parameters, Antioxidant Capacity, and Antimicrobial Activity of Honeys from Tropical Forests of Colombia: *Apis mellifera* and *Melipona eburnea*

**DOI:** 10.3390/foods12051001

**Published:** 2023-02-27

**Authors:** Isabel Cristina Zapata-Vahos, Juan Camilo Henao-Rojas, Diana Paola Yepes-Betancur, Daniela Marín-Henao, Carlos Eduardo Giraldo Sánchez, Samir Julián Calvo-Cardona, Dorely David, Mario Quijano-Abril

**Affiliations:** 1Facultad de Ciencias de la Salud, Grupo de Investigación Atención Primaria en Salud, Universidad Católica de Oriente, Rionegro 054040, Colombia; 2Corporación Colombiana de Investigación Agropecuaria–Agrosavia, Centro de Investigación La Selva, Kilómetro 7, Vía a Las Palmas, Vereda Llanogrande, Rionegro 054048, Colombia; 3Servicio Nacional de Aprendizaje, SENA, Grupo de Investigación en Innovación y Agroindustria, Centro de la Innovación, la Agroindustria y la Aviación, Cra 48 # 49-62, Rionegro 054040, Colombia; 4Grupo de Investigación en Estudios Florísticos, Facultad de Ingenierías, Herbario, Universidad Católica de Oriente, Rionegro 054040, Colombia; 5Grupo de Investigación Sanidad Vegetal, Facultad de Ciencias Agropecuarias, Universidad Católica de Oriente, Rionegro 054040, Colombia; 6Zootecnista, PhD Data Plus CIC, Pereira 660008, Colombia; 7Tecnoparque Nodo Rionegro, Centro de la Innovación La Agroindustria y la Aviación, Servicio Nacional de Aprendizaje—SENA, Rionegro 054040, Colombia

**Keywords:** bee honey, antioxidant potential, inhibitory activity, physicochemical property

## Abstract

Honey is a functional food used worldwide and recognized for its multiple health benefits. In the present study, the physicochemical and antioxidant properties of honey produced by two species of bees (*Melipona eburnea* and *Apis mellifera*) in two seasons were evaluated. In addition, the antimicrobial activity of honey against three bacterial strains was studied. The quality of honey analyzed by LDA (linear discriminant analysis) showed four clusters mediated by the interaction, the bee species, and the collection season resulting from a multivariate function of discrimination. The physicochemical properties of the honey produced by *A. mellifera* met the requirements of the Codex Alimentarius, while the *M. eburnea* honey had moisture values outside the established ranges of the Codex. Antioxidant activity was higher in the honey of *A. mellifera*, and both kinds of honey showed inhibitory activity *against S. typhimurium* ATCC 14028 and *L. monocytogenes* ATCC 9118. *E. coli* ATCC 25922 showed resistance to the analyzed honey.

## 1. Introduction

Honey is a sweet and viscous fluid produced by bees, which is one of the oldest traditional medicines and has had different uses in the culinary industry as a flavoring and sweetener. In addition, multiple beneficial health effects have been reported, such as treatments for burns, wounds, ulcers, gastrointestinal disorders, respiratory diseases, and cancer [1]. Some of these benefits are attributed to bactericidal and antioxidant honey activity [2].

The antioxidant and antibacterial activity of honey may be due to the presence of enzymes, such as glucose oxidase and catalase, as well as compounds, such as phenolic acids, flavonoids, organic acids, etc. [3]. Honey also contains macronutrients; carbohydrates represented as sugars and proteins, vitamins, minerals, and others [4]. The composition of honey varies according to the botanical origin as the main sources where the bees collected the nectar possibly being monofloral or polyfloral [5]. Another variation of honey could be due to the species of bees that produces it. *Apis mellifera* is a bee introduced from Europe. It belongs to the order Hymenoptera and family *Apidae*, and *Melipona eburnea* is a native South American stingless bee. It belongs to the same order and family *Meliponinae* [6]. However, few studies have compared the characteristics of the honey produced by these two species.

The quality of honey may vary depending on the species that produce it. Several parameters, such as moisture content, pH, acidity, organic acids content, and 5-hydroxymethyl- furfural (5-HMF) content determine the quality of honey [4]. Some research suggests the quality and physicochemical characteristics of honey could be better when hives are established at the limits of conserved natural forests in contrast to being located in areas close to urban centers [7]. The quality and other parameters of honey could be influenced by season [8,9]. In Colombia, beekeepers have not yet managed to take advantage of the high potential of the honey market due to the poor characterization and differentiation of their products and the high levels of counterfeiting with sugar [10]. Due to the megadiversity of flora that exists in the forests of Colombia [11], there is currently no representative sampling that allows us to elucidate the influence of its physicochemical characteristics.

The objective of this work was to characterize and compare the physicochemical properties and antioxidant and antimicrobiological activity of the honey produced by two species of bees, one native (*M. eburnea*) and the other introduced (*A. mellifera*) in the forests of the western slope of Magdalena in Colombia, which is considered a diversity hotspot.

## 2. Materials and Methods

### 2.1. Study Area

The study was carried out with 24 hives (12 from *Apis mellifera* and 12 from *Melipona eburnea*) established in two municipalities of eastern Antioquia, located in the very humid premontane forest life zone (El Carmen de Viboral and San Carlos, Antioquia, Colombia) (Figure 1). The selected municipalities are characterized by their current beekeeping production as a strategy for the generation of economic resources for communities in post-conflict times as well as forest conservation and management.

### 2.2. Obtaining the Samples

To have a better representation in terms of flowering peaks, climatic season, and harvest time, samplings were carried out at two times of the year (dry season and rainy season) [13]. For both *A. mellifera* and *M. eburnea*, the honey was obtained directly from the hives; collections involved between 15 and 20 mL of immature honey. The *M. eburnea* honey was collected using sterile 20 mL syringes where the samples of both kinds of honey were subsequently filtered and stored in plastic containers marked with the hive code and the date of collection. Nine (9) experimental replicates were taken for each possible interaction between bee species, season, and municipality, while each experimental variable was measured in triplicate in the laboratory for each real sample.

### 2.3. Physicochemical Characterization

The physicochemical parameters of the honey of the two bee species were evaluated following the described methods in Codex Alimentarius [14] and official analytic methods of the AOAC [15]. The moisture content (% *w*/*w*) was determined by the refractometric method using a refractometer (ATAGO), and it was calculated using the Wedmore table. The pH values were measured with a pH meter (Jenway) using a solution of 10 g of honey in 75 mL of distilled water. Free and lactonic acidity were determined potentiometrically by titration, adding NaOH (0.05 mol/L) until reaching pH of 8.3 and 8.5, respectively (mg equivalent of acid. Kg^−1^). The ashes (%) were analyzed by muffle incineration, achieving constant weight. Electrical conductivity was measured according to the method proposed by the International Honey Commission of 2009 (mS/cm). The color was measured by the Bianchi method combining the color parameters determined in mm of Pfund with the absorbance of samples at a given wavelength.

#### 2.3.1. Diastase and HMF Activity

Diastase activity was determined using the AOAC 958.09 method. A buffered mixture of soluble starch and a honey solution was used in a water bath at (40 ± 1 °C) for the time required to reach a specified endpoint (determined spectrophotometrically) until reaching an absorbance value (0.235) at 660 nm (Diastase Number, DN).

#### 2.3.2. HMF Activity

The HMF content determination was based on the method by Zappala et al. (2005). Amounts of 3.5 g of honey samples were diluted up to 5 mL with distilled water, filtered on 0.45 mm filter, and immediately injected in a UHPLC (UltiMateTM 3000 high-performance liquid ultrachromatography equipment) equipped with a diode array detector. The HPLC column was a Raptor C18, 2.7 μm, 150 × 3 mm. The HPLC conditions were the following: isocratic mobile phase, 90% water at 10% methanol; flow rate, 0.4 mL/min; and injection volume, 10 μL. All the solvents were HPLC grade (Merck, Milan, Italy). The wavelength range was 220–660 nm, and the chromatograms were monitored at 283 nm [16].

#### 2.3.3. Sugars

Three types of sugar were analyzed, fructose, glucose, and sucrose, for the two kinds of honey produced by the two species of bees. UHPLC UltiMateTM 3000 high-performance liquid ultrachromatography equipment was used. A total of 3.5 g of honey was weighed and diluted with distilled water to 5 mL; then, 18 µL was taken and made up to 25 mL in water for an acetonitrile mixture (50:50) and passed through 0.22 µm syringe filters. A CAD aerosol charge detector was used with an acetonitrile solution. Water (75:25) as a mobile phase and a column Asahipak 5 µm NH2p-50 4E 100ª of 250 × 4.6 mm and a flow of 1 mL/min was used. The oven temperature was 30 °C. The values are expressed in percentages (%).

### 2.4. Determination of Total Phenols Content

The determination of phenols was carried out by the Folin–Ciocalteu colorimetric method [13]. In a reaction tube, 50 µL of the honey dilution, 425 µL of distilled water, and 125 µL of the Folin–Ciocalteu reagent were added. This was stirred and then allowed to stand for 6 min. Subsequently, 400 µL of 7.1% Na_2_CO_3_ was added. After 1 h in the dark, absorbance was read at 760 nm. A standard curve was constructed using gallic acid as a standard. The analyses were carried out in triplicate, and the results were expressed as mg equivalent of gallic acid/100 g of honey.

### 2.5. Antioxidant Capacity Analysis

#### 2.5.1. DPPH Free Radical Trapping Activity

The Brand-Williams method was used with some modifications [17]. In a test tube, 10 µL of the honey dilution and 990 µL of a DPPH solution were added. As a reference, the same amount of DPPH and 10 µL of the sample solvent (water) were used. The antioxidant capacity was evaluated using the decrease in absorbance after 30 min of reaction at a wavelength of 517 nm. The percentage of radical inhibition was calculated, and the results were expressed as TEAC (Trolox equivalent antioxidant capacity) values (μmol of Trolox/100 g of honey) by constructing a standard curve, using Trolox as an antioxidant. The analyses were carried out in triplicate.

#### 2.5.2. ABTS Free-Radical-Trapping Activity

Evaluation of the antioxidant capacity by the ABTS+ cationic radical method was used [12]. The radical was activated by a reaction between ABTS with potassium persulfate. In a test tube, 10 µL of the honey dilution and 990 µL of an ABTS solution standard at 0.7 of absorbance were added. As a reference, the same amount of ABTS and 10 µL of buffer 7.4 were used. The ability of the samples to trap the ABTS radical was evaluated using the decrease in absorbance after 30 min of reaction and at 732 nm of wavelength. The results were expressed as TEAC values by constructing a standard curve using Trolox^®^ as an antioxidant.

#### 2.5.3. FRAP (Reducing Capacity)

FRAP was carried out according to the method of Benzie and Strain [18]. A volume of 50 μL of honey was mixed with 950 μL of solution FRAP (A solution of TPTZ, FeCl_3_, and acetate buffer pH 3.6). After 30 min, absorbance data were measured at a wavelength of 593 nm. The reference curve was constructed using ascorbic acid. The activities of the samples were expressed as AEAC (antioxidant capacity in ascorbic acid equivalents: mg of ascorbic acid/100 g of honey).

### 2.6. Antimicrobial Activity

The antimicrobial activities of the different kinds of honey were determined using the diffusion methodology in three bacterial strains, *Escherichia coli* ATCC 25922, *Salmonella typhimurium* ATCC 14028, *Listeria monocytogenes* ATCC 19118, which were selected for their relationships with spoiled or contaminated food. This methodology makes it possible to measure the area of inhibition of the two kinds of honey. The microorganisms were reactivated 24 h before on TSA agar (trypticase soy agar) seeded by the exhaustion method and incubated at 37 °C. They were then inoculated in BHI (brain heart infusion) until reaching turbidity equivalent to the concentration of 0.5 of the McFarland standard. This solution was massively seeded on the surface of the Muller Hinton agar using a sterile swab and making a horizontal sweep along the entire surface. The process was repeated three times. Rotating the petri dish 60°, 0.5 mm discs were impregnated with each of the kinds of honey, and they were deposited in the culture medium in an equidistant and random manner. Three repetitions of each of the evaluated honey were made and a solution of ciprofloxacin (160 mg/mL) was used as a positive control [19,20].

### 2.7. Statistic Analysis



**Determination of Data Normality Criteria**



Descriptive statistical analyses were obtained for each of the variables evaluated, and hypothesis tests were also applied to meet the assumptions of normality and homoscedasticity using the Shapiro–Wilk and Bartlett tests, respectively [21]. Additionally, the multivariate normality test was performed, calculating Royston’s H index [22]. This was done both in the original data and in the normalized matrices, confirming the effectiveness of the process.



**Linear Discriminant Analysis (LDA) and Multiple Range Tests**



To determine the influence of the factors on the response variables, given the impossibility of performing an analysis of variance due to the non-parametric nature of the data, a linear discriminant analysis was carried out using all possible combinations of the controlled factors as discriminant factors (honey collection season, bee species, and their interaction). To perform this analysis, it was necessary to transform the data through a standardization process using the Z-score methodology reported by Kappal [23], establishing linear polynomial relationships between all the variables studied to differentiate between categorical groups that were previously defined as well as their interactions. In this case, to establish which physicochemical, functional, and microbiological variables of the kinds of honey had discriminant weight, the Lambda-Wilks parameter was used [24], which was chosen to verify that it should not be greater than 0.05. Subsequently, each of the clusters formed for each variable was compared using the non-parametric Kruskal–Wallis test (*p* < 0.05). It is important to highlight that this second stage of analysis should have been carried out with the original database (un-normalized) to preserve the dimensionality and interpretation of the measurements.



**Principal Component Analysis (PCA) and Correlation Analysis**



To reduce the dimensionality and determine the variables that contribute the most to the general variability of the honey population studied, a principal component analysis (PCA) was used for the variables associated with the quality of the honey of each species of bee, which allows observing the behavior of the variables characterized as a whole for each type of honey [25]. In addition, the correlation coefficient matrix (r) was constructed with Spearman method using the Factoextra [26] and corrplot libraries [27]. All the aforementioned analyses were carried out using the statistical software, Statgraphics Centurion XVI.II, in the company of the free software, R, and its R Studio complement.

## 3. Results and Discussion

### 3.1. Influence of the Bee Species and the Time of Collection on the Quality Characteristics and Biological Activities of the Two Kinds of Honey Produced in Tropical Forests of Colombia

#### 3.1.1. Physical–Chemical Analysis

Figure 2 shows the proposed linear discriminant model represents 96.95 % of the total variability of the sampled population (LD1:54.31%, LD2: 42.64%). In addition, it is possible to observe the formation of clusters according to the species of bees from which the honey comes and the honey collection season. There are four clearly defined clusters, which confirms that honey quality is mediated by the interaction, the bee species, and the collection season resulting from a multivariate function of discrimination (Figure 2b). On the other hand, the municipality did not have an appreciable discriminating capacity. These clustering results are like those obtained by Conti [28], although they found the grouping based on the quality characteristics of the honey according to their locality of production and could not demonstrate a statistical relationship of such clustering. It is important to highlight the variables with the greatest discriminating power between the honey of two species of bees and in two collecting seasons are: D-fructose, D-glucose, D-sucrose, ABTS radical uptake capacity, total polyphenolic compounds, electrical conductivity, and free acidity.

The multiple range test of Kruskall–Wallis *p*-values for the characteristics in the honey produced by the two species of bees during the two collect seasons are shown in Table 1.

The color variable did not show normal behavior, and the Kruskal–Wallis test did not show significant effects between the honey produced by *A. mellifera* and *M. eburnea* bees (*p* ≥ 0.05), and there were also no differences by season. According to the Pfund scale to classify the color of honey [29], the honey produced by *A. mellifera* and *M. eburnea* correspond to light amber in the dry season (72 and 60.83 mm PFund, respectively) and extra light amber in the rainy season (48.21 and 49.69 mm PFund, respectively). Similarly, other researchers have found variations in the color of honey are related to the floral source and the collection season [30,31] since the heat could exert a darkening action; for example, Melaku and Tefera [32] found *A. mellifera* honey is darker in the May to June harvest season (equivalent to a dry season) than the honey harvested in the October to November season (equivalent to a relatively rainy season). In the honey stingless bee (Tetragonula species), a Pfund average of 67 ± 19 mm, which classifies it in the color range of light amber [33], was reported.

The honey of the two species fluctuated between 0.15 and 0.26% ash. There were no significant effects for the ash content in the honey produced by the studied species (*p* ≥ 0.05) nor by season. The ash contents of honey were higher than those reported by Melaku and Tefera [32], who found an average ash content of 0.12 ± 0.12% in A. mellifera honey samples, while the honey from different species of Mellipona was reported with values higher than those found in the present study [34]. According to Nanda et al. [35], mineral content of honey is highly dependent on the types of flowers used by bees for nectar.

The species *A. mellifera* had moisture values lower than those reported by *M. eburnea* (*p* ≤ 0.05); this variable showed no variation by season. The moisture content of honey is an important factor reflected in its shelf life since honey with high moisture content (above 20 %) can be susceptible to fermentation during storage [36]. Likewise, honey with a moisture content between 18 and 20% is considered mature and stable [14]. The moisture content of the honey fluctuated between 19.8 and 26.9%, which may be related to the high relative humidity of the municipalities where the study was carried out, which has also been reported by Adgaba et al. [37] who suggested high-moisture content of honey from humid regions can be related to the difficulties of bees to evaporate moisture from honey against high relative humidity in the air.

The pH of honey from *A. mellifera* had significant differences according to the season (*p* ≤ 0.05), but not between species. The pH values in the honey were between 3.55 and 4.48, which is similar to the average pH value of 3.02–4.16 found in a study from *A. mellifera* honey in Eastern Amhara Region, Ethiopia [32]. The free acidity level of honey samples ranged from 36.12 to 49.46 meq acid. Kg^−1^ and did not present statistically significant differences between the species nor the temporality (*p* ≥ 0.05), whereas lactonic acidity in honey showed differences mediated by season but not by bee species. In the same way as our study, the honey produced by *M. favosa* also found a high average free acidity of 50.6 meq acid. Kg^−1^, indicating the presence of higher amounts of weak acids, such as organic acids with low ionization [38]. According to Nascimento et al. [39] and Apriceno et al. [40], honey acidity depends on its content of organic acids, particularly the gluconic acid that results from the spontaneous hydrolysis of glucone-δ-lactone enzymatically formed by glucose. Therefore, the total acidity in honey must be evaluated as the sum of free and lactone acidities; likewise, the lactonic acidity is considered the reserve of acidity when the honey becomes alkaline [41].

Regarding the HMF analysis, significant differences were observed mainly for the rainy collection season between the two species of bees. The values found are low considering the limits established by the Codex Alimentarius (less than 80 mg/Kg) [30], which further suggests the freshness of the honey. Other researchers have reported values higher than those found in the present study; for example, Hoxha et al. [42] found HMF values between 12.61 and 663.58 mg/kg in local and imported brand honey available in different markets. Likewise, in 21 Italian honey samples, the HMF values ranged from 3.35 to 43.21 mg/kg [34]. The 5-Hydroxymethylfurfural (HMF) is a cyclic aldehyde formed during the decomposition of fructose and glucose. However, factors, such as acidic conditions, high temperature, high water content, and metallic containers, can influence their formation; likewise, it can be produced during food processing through the Maillard reaction or extended storage [32,42,43]. Consequently, High HMF levels generally provide an indication of overheating, storage in poor conditions, or aging of the honey. In addition, HMF can be converted to 5-sulfoxymethylfurfural (SMF), a genotoxic compound in in vivo conditions, which represents a risk to the health of consumers [41,44].

The diastase activity values for the honey produced by the species *A. mellifera* were 12 and 60 diastase numbers for the dry and rainy collection seasons, respectively, while the honey produced by the *M. eburnea* did not show activity (results are not included in the table). Diastase activity in honey is related to its freshness and heat treatment; this activity also can vary depending on the floral sources used by bees. In honey regulatory standards, it has been established that diastase activity should not be less than eight diastase number (DN) units, where 1 DN unit hydrolyses 1 mL of 1 % starch solution using 1 g of honey for 1 h at 40 °C [43,45]. In monofloral and polyfloral honey of *A. mellifera* from different regions of Brazil, similar values of diastase activity have been reported, varying between 7.15 and 57.69 DN [39]. In other studies, very low diastase activity has also been found in honey from the Melipona bee genus, which could be part of its nature [33,38,46]; this suggests this parameter should not be indicative of the quality of honey from *M. eburnea*.

Glucose and fructose presented statistically significant differences by species and by season (*p* ≤ 0.05); values in the dry season were higher than in the rainy season. Regarding sucrose, there were differences between *A. mellifera* in the dry season and *M. eburnea* in the rainy season, presenting a high consideration in the values obtained in *A. mellifera* in the rainy season and *M. eburnea* in the dry season. The soluble solids (°Brix) did not appear different between *A. mellifera* and *M. eburnea* in the rainy collection season, while different for *A. mellifera* in the dry season (*p* ≥ 0.05). The sugar values reported in this research showed similar behavior to those shown by Nascimento et al. [39] who reported fructose can represent 36% and glucose 31%. Honey crystallization depends on the fructose-to-glucose ratio (F/G) since glucose is less soluble in water than fructose, and higher values of 1.14 of the fructose/glucose ratio indicate honey tends to crystallize easily [47]. In this study, this relationship varied between 1.17 and 1.19 for *M. eburnea* and *A. mellifera* from 1.18 to 1.25 according to the collected season (dry and rainy, respectively). The average sucrose content of honey samples ranged from 0.19 to 2.43%, which is slightly lower than that reported by Melaku et al. [32]. In addition, a low honey sucrose concentration could be related to it being fully converted into glucose and fructose by the action of invertase enzyme [48].

The electrical conductivity did not present differences between the values obtained by the season from *A. mellifera* and *M. eburnea*, but there were differences in the electrical conductivity of honey from *M. eburnea* in the dry and rainy season (152 and 126.8 mS/cm, respectively). The values in the present study are lower than those shown by [35] who reported values between 173 to 927.33 mS/cm in multi-floral honey produced in some regions of Algeria. Honey electrical conductivity is strongly dependent on the concentrations of mineral salts and organic acids, and this parameter could show variability according to the floral origin and is a factor integrated into the international standards of honey for the discrimination of honeydew and honey blossom [36,49].

The results of the physicochemical parameters for the two classes of honey are within those established by the Codex Alimentarius [14]: ashes less than 0.6%, HMF less than 80 mg/Kg, the sum of fructose and glucose greater than 45 g. 100 g^−1^, sucrose less than 5 g. 100 g^−1^, and diastase greater than 8 Schade units. However, parameters, such as humidity and acidity, were higher than the values required by the standard (20% and 40 meq acid. Kg^−1^, respectively) for the honey produced by the species *M. eburnea*. This indicates the honey produced by species without a sting might have a shorter shelf life due to a tendency for fermentation. Studies, such as [34], suggest a normative basis for honey produced by stingless bees, showing higher values in acidity and humidity compared to honey produced by *A. mellifera*. This is reaffirmed by studies in Brazil [6].

The differences in the physicochemical parameters presented between the dry and rainy seasons could be explained by environmental factors, nectar moisture content, time of year, place of collection, soil composition, and degree of maturity of the honey [50]. Likewise, it is important to note not all the plant species selected by the bees showed constant blooms throughout the year; therefore, the resource was not always the same [13].

#### 3.1.2. Biological Analysis

Table 1 shows the results of antioxidant activity and total phenol compounds. Low DPPH radical-scavenging activity was found for both A. Mellifera and *M. eburnea*, showing a higher antioxidant capacity for *A. mellifera* honey (*p* ≤ 0.05). In addition, differences were found according to the harvest rainy season between the two species. On the other hand, ABTS radical-scavenging activity presented differences between species in the dry season.

The honey produced by *A. mellifera* showed a greater reducing capacity by the FRAP method than the honey produced by *M. eburnea* (*p* ≤ 0.05). This variable did not show significant differences between the honey collection seasons. Honey from A. Mellifera had a higher content of total phenols compared to honey from *M. eburnea* but did not show differences for the collection season.

Considering that antioxidants act via multiple mechanisms depending on the reaction system or the radical or oxidant source, the antioxidant activity was measured by three methods (ABTS, DPPH, and FRAP) [51].

The results obtained in this study are similar to those obtained by Nascimento [39] who report a range between 26.0 and 100.0 mg GAE100 g^−1^ for monofloral and multi-floral honey produced by A. mellifera in Brazil. Alvarez Suarez et al. [52] reported a value of 54.30 mg GAE. 100 g^−1^ for polyfloral honey. The phenol content in honey is an important parameter that not only determines the quality but also its biological potential [49,53], mainly as antioxidant activity [54]. The amount of phenols found in the present analysis provides added value to the honey produced in this type of forest ecosystem.

On the other hand, the ability to trap free radicals of the honey (studied here by DPPH and ABTS) was lower compared to other studies carried out in Mozambique on commercial honey that had values of 19.17 and 160.01 µmol of Trolox 100 g^−1^ for DPPH [55]. Bodó et al. [56] found ABTS values of 100 µmol of Trolox. 100 g^−1^ in multi-floral honey from Hungary. This adds up to an extraordinary number of biological properties attributed to honey since it has anti-inflammatory, antibacterial, antiviral, and anticancer effects. Some compounds found in honey and related to health effects are polyphenols compounds, such as vanillic acid, caffeic acid, ellagic acid, syringic acid, ferulic acid, *p*-cumaric acid, benzoic acid, and others; flavonoids, such as apigenin, quercetin, pinocembrin, chrysin, galangin, and kaempferol; and antioxidants, such as tocopherols, ascorbic acid, superoxide dismutase, and catalase [49,53].

The inhibition capacity of honey against some microbial agents, such as *S. typhimurium* and *L. monocytogenes,* can be seen in Table 1. The results relating to the inhibition capacity of the honey against the bacterium, *E. coli* ATCC 25922, suggest resistance of the strain against the evaluated samples. The results obtained showed both species, *A. mellífera* and *M. eburnea,* exhibited inhibition capacity for the bacteria, *S. typhimurium* and *L. monocytogenes*. The last one showed a greater inhibition halo and presented a statistical significance between A. Mellifera in the dry and rainy seasons (*p* ≤ 0.05); *M. eburnea* did not exhibit differences. *S. typhimurium* did not present differences between the species or collection season (*p* ≥ 0.05)

The results of antimicrobial activity for the two types of honey in this study showed there is no activity against the *E. coli* ATCC 25922 strain that is consistent with that reported by Aguilera et al. [57] in A. Mellifera honey. In contrast, [58] found antimicrobial activity against the *E. coli* strain ATCC 31617 for the angel bee species (Tetragonisca angustula).

The two kinds of honey in the present study showed inhibitory activity against *L. monocytogenes* and *S. typhimurium* strains. Similar studies found an inhibitory effect of multi-floral honey against *L. monocytogenes* 1/2B and *S. typhimurium* NRRLE 4463 at 50 and 75%, respectively [59]. Some authors attribute the antibacterial activity of honey to the presence of hydrogen peroxide resulting from the activity of catalase and glucose oxidase as well as the presence of phenols compounds, phenolic acids, flavonoids, ascorbic acid, organic acids, methylglyoxal, and bee protein defensin 1 [3,60]. The variation in antimicrobial activities of honey mainly depends on the flower types, environmental conditions, geographical location of the floral sources, peroxide activity, and non-peroxide mechanisms that result in different bioactive compounds [61,62].

### 3.2. Principal Components Analysis

The results of the principal component analysis (PCA) are consolidated in Figure 3, where two independent data structures are shown given the grouping between the two bee species, obtaining a representation of the variability of the samples of 78.75 and 82.18% for *A. mellifera* and *M. eburnea*, respectively (Figure 3a,b).

Figure 3a shows in the case of *A. mellifera*, the main component one is associated with the mineral content of honey, antioxidant capacity, and total polyphenol content. Component 2 is mainly associated with the °Brix content and humidity. Additionally, it can be observed the variables that contribute in greater proportion to the variability both in factors associated with quality and in their biological activities of *A. mellifera* honey are D(+)-saccharose, °Brix, HMF, humidity, and inhibitory capacity of Listeria monocytogenes. These results are like those found by Conti et al. [28] where it is shown that in a multivariate way, humidity and sugar content accompanied by some minerals, such as magnesium and potassium, are quality variables that contribute a greater proportion to the variability of South American unifloral and multi-floral honey, which is directly related to the characteristics of the polem minerals that contain the local floral species.

Regarding the PCA obtained for the honey collected from *M. eburnea* (Figure 3b), it is observed that component one is highly related to the DPPH radical-trapping activity, pH, and electrical conductivity, these variables being the ones that contribute most to the variability of the samples. For its part, component two is related to the inhibitory activity of Listeria monocytogenes and the scavenging capacity of ABTS radicals. For this species, the variables that contribute in greater proportion to the variability of quality are related to pH, electrical conductivity, and the capacity to capture radicals DPPH and ABTS accompanied by the content of total polyphenols. This indicates high interspecific variability in the functional properties of the honey produced by stingless bees, which could be explained by the incipient research on quality improvement regarding the homogeneity, stability, and biochemical properties of the honey produced by stingless bees, accompanied by its lack of control for its sale and incipient regulations in force worldwide, as reported by Braghini et al. [63]. Additionally, some probable correlations are appreciated given the closeness of the vectors for both kinds of honey, a hypothesis that was confirmed by the analysis shown in the following section.

### 3.3. Correlations Analysis

The analysis of independent correlations between the variables studied in *A. Mellifera* and *M. eburnea* honey are shown in Figure 4a,b, respectively. In the case of honey produced by *A. Mellifera*, free acidity had a significant direct relationship with lactonic acidity and ash. It is like Mulugeta et al. [64], and free acid had a negative correlation with Brix and Sucrose. In *M. eburnea*, this variable was directly related to HMF, moisture, lactonic acidity, and ABTS and negatively related to DPPH, Brix, and pH. The last one was according to other authors. It is due to the organic acid, such as gluconic acid, that increases ion H+ and decreases pH [65].

Regarding color, honey obtained from *A. mellifera* had positive relationships with TPC, FRAP, minerals, HMF, free acidity, and lactonic acidity and a negative relationship with sucrose. Honey from *M. eburnea* had positive relationships with TPC, minerals, glucose, and pH and negative correlations with sucrose, electrical conductivity, and *S. typhimurium* inhibition. Some authors report a high correlation between color, FRAP, and phenols. It could be explained by the color of honey being influenced by pigments, phenols, minerals, HMF, and products of the Maillard reaction, and they present antioxidant activity [4,66].

Sucrose concentration for *A. mellifera* showed positive relationships with fructose and pH and negative relationships with ABTS, free acidity, TPC, lactonic acidity, color, and Listeria inhibition. For *M. eburnea* honey, sucrose had only positive correlations with Brix and negative correlations with TPC, ABTS, color and glucose, and free acidity. Electrical conductivity showed positive correlations with fructose and HMF. Negative correlations are presented with DPPH and lactonic acidity. The electrical conductivity of *M. eburnea* showed positive correlations with free acidity and HMF. Negative correlations were found with DPPH, pH, TPC, ABTS, color, ash, and FRAP. The measurement of electrical conductivity depends on the free acid of the honey. The higher the acid content, the higher the resulting conductivity. This is in agreement with Mulugeta et al. [64].

In the case of the *S. typhimurium* inhibition variable for A. Mellifera, it did not show statistically significant relationships with any measured variable, and in the case of *M. eburnea* honey, it showed negative correlations with color and Brix. The ABTS variable for A. Mellifera honey showed positive correlations with FRAP, TPC, and Listeria inhibition and negative correlations with sucrose, fructose, and pH. Regarding ABTS correlations for *M. eburnea* honey, it showed positive correlations with TPC, ash, moisture, lactonic acidity, free acidity, and HMF and negative correlations with electrical conductivity, sucrose, fructose, and *L. monocytogenes* inhibition.

There was high variability in the results and a high correlation of phenols with FRAP, ABTS, and DPPH, which explains why the antioxidant activity was due to the presence of secondary metabolites, such as phenolic compounds. These phenolic compounds come from nectar sources, which varied throughout the evaluated seasons, as well as the diversity of plants visited by bees [8]. Results showed higher levels of these phytochemicals could produce a more potent antibacterial effect against *L. monocytogenes*. Some authors showed there is a high correlation between phenolic compounds, such as quercetin, rutin, and chlorogenic acid, with antibacterial activity, showing greater inhibition in Gram-positive bacteria than in Gram-negative bacteria [3].

## 4. Conclusions

Bee species and the collection season influence the quality characteristics of the different kinds of honey produced by *A. mellifera* and *M. eburnea* in tropical forests of Colombia.

The honey produced by *A. mellifera* complies with the Codex Alimentarius parameters, and those produced by *M. eburnea* have some parameters out of range; thus, this study proposes the construction of a different standard for honey produced by stingless bees than for native bees. In addition, the antioxidant activity had higher values in honey from *A. mellifera*. Honey inhibited the growth of the two strains, *L. monocytogenes* and *S. typhimurium*. The inhibition halo was greater in *L. monocytogenes*.

The establishment of hives for productive purposes inside these forests can be an important strategy due to the high diversity of resources for bees as well as the frequency of their availability. Later studies focusing on the characterization of this type of phenol can elucidate the chemical diversity of this type of honey and its biological potential.

## Figures and Tables

**Figure 1 foods-12-01001-f001:**
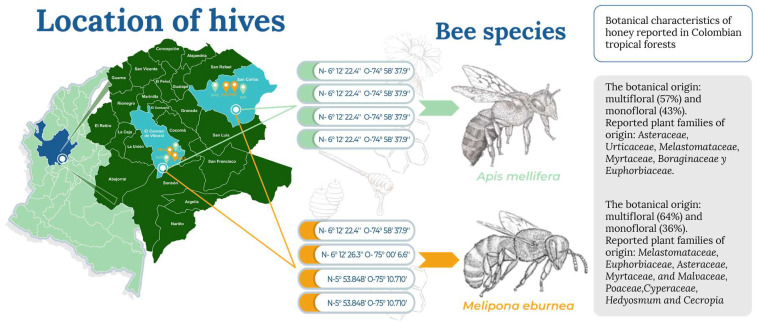
Location of sampled hives, morphological illustrations, and botanical characteristics of the honeys produced by *A. mellifera* and *M. eburnea* reported in tropical forests of Colombia. Botanical characterization of honey adapted from [12].

**Figure 2 foods-12-01001-f002:**
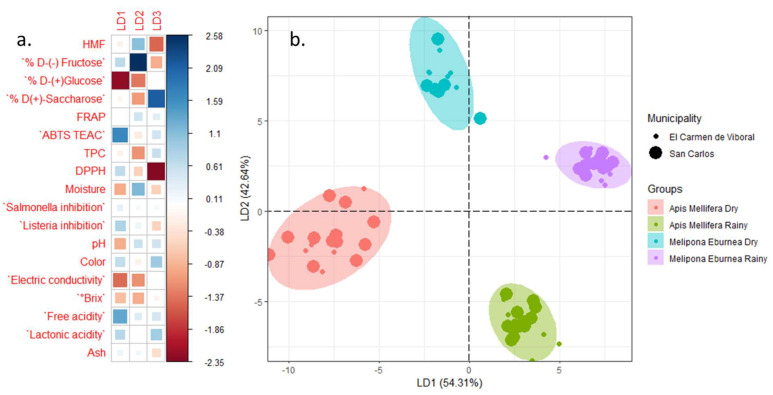
Linear discriminant analysis for honey produced by *Apis mellifera* and *Melipona eburnea* in humid and dry harvesting seasons of tropical Colombian forest. (**a**) Discriminant weight of the variables associated with quality. Larger size of the squares and more color intensity indicates higher discriminant weight. (**b**) Cluster plot showing clustering of honey by the interaction of bee species and honey harvesting season.

**Figure 3 foods-12-01001-f003:**
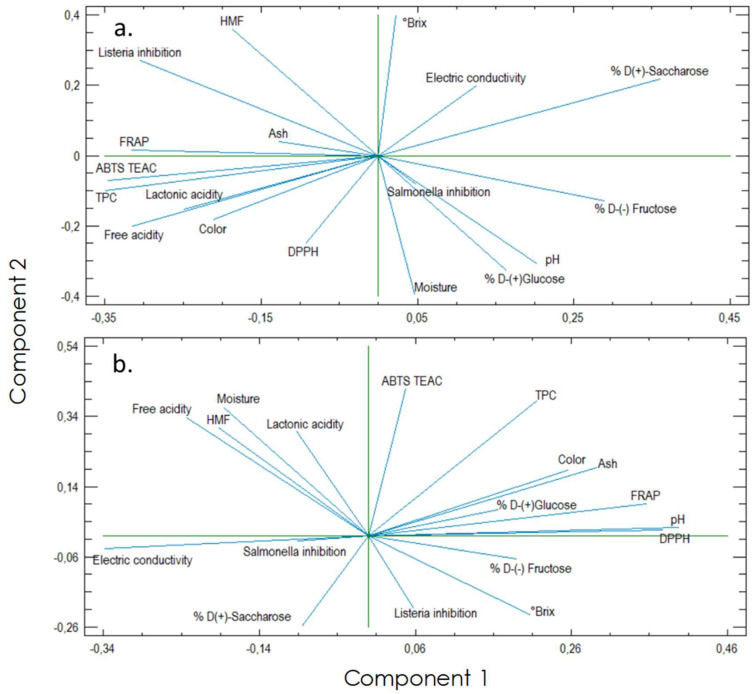
Principal component analysis of variables associated with the quality of honey produced by (**a**) *Apis mellifera* and (**b**) *Melipona eburnea* in Colombian forest.

**Figure 4 foods-12-01001-f004:**
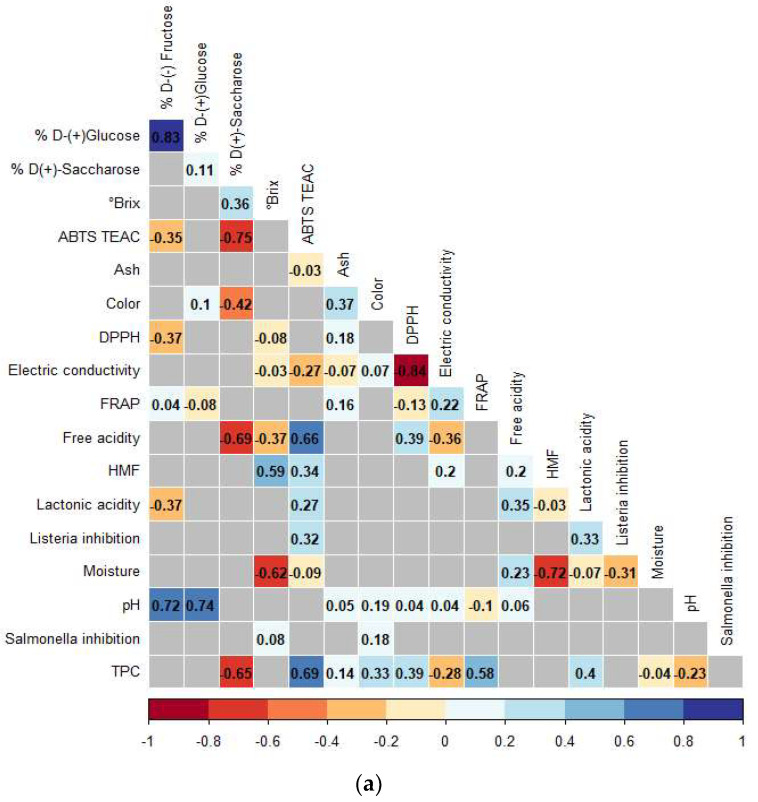
Spearman’s correlation analysis between the variables associated with the quality of honey produced by (**a**) *Apis mellifera* and (**b**) *Melipona eburnea* in tropical Colombian forest. he presence of color indicates statistically significant relationships (*p* < 0.05), and the intensity of red or blue color indicates the magnitude of the correlation coefficient, whether it is an inversely proportional or directly proportional relationship, respectively.

**Table 1 foods-12-01001-t001:** Comparison of multiple ranges of Kruskall–Wallis for variables associated with the quality of honey produced by *Apis mellifera* and *Melipona eburnea* in the dry and rainy seasons in the Colombian Andean region. Different letters indicate significant differences between treatments according to the Kruskall–Wallis test.

* Variable *	* A. mellifera * × Dry	* A. mellifera * × Rainy	* M. eburnea * × Dry	* M. eburnea * × Rainy	*p*-Value KW Test
Color (mm PFund)	72 ± 21.2 a	48.21 ± 24.8 a	60.83 ± 21.7 a	49.69 ± 23.2 a	0.186
Ash (%)	0.2 ± 0.09 a	0.26 ± 0.14 a	0.155 ± 0.02 a	0.15 ± 0.1 a	0.079
Moisture (%)	22.68 ± 2.9 a	19.8 ± 2.6 a	25.44 ± 1.7 b	26.94 ± 2.2 b	0.000
pH	4.15 ± 0.38 b	3.55 ± 0.23 a	4.48 ± 0.8 ab	3.67 ± 0.24 a	0.000
Free acidity (meq acid. Kg^−1^)	42.14 ± 7.4 a	46.47 ± 7.01 a	36.12 ± 27.3 a	49.46 ± 25.2 a	0.786
Lactonic acidity (meq acid. Kg^−1^)	0 ± 0.01 a	0.2 ± 0.06 b	0 ± 0.02 a	0.5 ± 0.05 b	0.000
HMF (mg/Kg)	4.82 ± 2.46 ab	7.42 ± 2.19 b	4.34 ± 2.7 ab	3.8 ± 3.3 a	0.032
% D-(+) Glucose	30.11 ± 2.41 b	22.19 ± 4.34 a	27.94 ± 1.37 b	20.7 ± 3.45 a	0.000
% D-(−) Fructose	36.63 ± 2.20 d	27.74 ± 1.31 c	32.94 ± 1.95 b	24.63 ± 2.52 a	0.000
% D(+)-Saccharose	2.73 ± 0.69 b	0.9 ± 0.84 ab	0.42 ± 0.4 ab	0.19 ± 0.16 a	0.000
°Brix	77.5 ± 1.94 b	77.8 ± 2.70 a	69.5 ± 2.23 ab	71.95 ± 1.63 a	0.007
Electric conductivity (mS/cm)	144.5 ± 21.89 ab	134.4 ± 12.12 a	152 ± 60.85 b	126.80 ± 15.40 a	0.007
DPPH (mmol Equiv TX/100 g)	0.76 ± 0.24 bc	1.05 ± 0.23 c	0.05 ± 0.26 ab	0.095 ± 0.02 a	0.000
ABTS TEAC (mmol Equiv TX/100 g)	4.12 ± 2.10 b	9.97 ± 5.05 b	0.36 ± 1.38 a	2.25 ± 1.6 ab	0.000
FRAP (mg Equiv ascorbic acid/100 g)	73.22 ± 28.50 b	117.46 ± 29.99 b	15.61 ± 14.25 ab	10.79 ± 2.69 a	0.000
TPC (mg Equiv GAE/100 g)	49.2 ± 6.94 b	62.39 ± 12.23 b	25.69 ± 9.44 a	32.89 ± 13.14 a	0.001
* S. typhimurium * inhibition (mm)	8.71 ± 0.85 a	8.66 ± 0.71 a	8.69 ± 1.03 a	8.29 ± 2.84 a	0.470
Listeria inhibition (mm)	10.49 ± 7.76 a	26.03 ± 4.91 b	25.56 ± 4.60 ab	23.26 ± 6.28 ab	0.015

## Data Availability

Not applicable.

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
