# Peer review of "Physicochemical Parameters, Antioxidant Capacity, and Antimicrobial Activity of Honeys from Tropical Forests of Colombia: Apis mellifera and Melipona eburnea"

_foods, 2023, doi:10.3390/foods12051001_

Round 1
Reviewer 1 Report
The paper needs thoughtful revision to be reconsidered by the journal.
Line1: Please replace “Type of the Paper (Article)” by “Article”
Line 145/154: FeCl3 - Na2CO3 :
Please place 2.5 Determination of total phenols content before section “ 2.4 Antioxidant Capacity”.
Please reupload higher quality figure 1.
Table 2: The different characterization methods and quantified compounds in honeys used in this study. is it based on Honey Quality and International Regulatory Standards!!? In whish basis the criteria were selected to compare honey samples.
Results should be presented as Mean±SD. How many replicates done for each analysis to confirm the obtained results!
Figure 2: PCA seems confusing, the Principal Component Analysis shows the parameters with plotting of the different honey samples. This cannot provide utile information for the study. The same for figures 3 the correlation analysis is done to show if there is a relation between parameters.
Overall, the paper do not focus deeply on honey sample characterization (effect of Saison or sample origin) however the illustrations presented to see the correlations between parameters.
Author Response
Response to reviewers
Manuscript ID: foods-2120784
Manuscript title: Physicochemical parameters, antioxidant capacity and antimicrobial activity of honeys from tropical forests of Colombia: Apis mellifera and Melipona eburnean
Dear Prof. Dr. Arun K. Bhunia
Editor-in-Chief
Foods
On behalf of all the co-authors of this document, I would like to thank the work and time of the jurors in reviewing in detail and contributing to the improvement of the quality and clarity of our work. We find the reviews helpful and have responded to each of your comments in detail. Based on the reviewer's comments, we made modifications to the original manuscript and carefully revised it.
We detail our responses to each of the comments below, and we believe that the manuscript has been greatly improved by your input. We hope that this contribution will be to the liking and approval of the reviewers for publication in their journal.
For the answers and modifications, we refer in each of them to the line numbers. In addition, the language of the document was thoroughly reviewed and corrected as requested, a process carried out by an English language expert.
Response to reviewer 1 comments:
- Line 1: Please replace “Type of the Paper (Article)” by “Article”
R: Reviewer's suggestion accepted and corrected in document text.
- Line 145/154: FeCl3 - Na2CO3:
R: The lines suggested by the reviewer were modified and the molecular formulas were corrected (Line 164/138).
- Please place 2.5 Determination of total phenols content before section “2.4 Antioxidant Capacity”.
R: The reviewer's suggestion is accepted, and a modification is made in the order of the methodological sections, remaining with numeral 2.4 Determination of total phenols content and 2.5 Antioxidant Capacity.
- Please reupload higher quality figure 1.
R: The reviewer's recommendation is accepted and figure 1 was reloaded in the document with the best image quality.
- Table 2: The different characterization methods and quantified compounds in honeys used in this study. is it based on Honey Quality and International Regulatory Standards!!? In whish basis the criteria were selected to compare honey samples.
R: The variables chosen in this document obey those referenced in the Codex Alimentarius version 2019, specifically in the International Standard for Honey CXS 12-19811 section. At the reviewer's suggestion, this information is supplemented on lines 99 to 101 and the respective reference is placed.
- Results should be presented as Mean±SD. How many replicates done for each analysis to confirm the obtained results!
R: The complement suggested by the reviewer is made and the standard deviation values are included in Table 2. Additionally, the following text related to the number of replicates and pseudoreplicates of the experimental design is added: Line 86 to 88: “Nine (9) experimental replicates were taken for each possible interaction between bee species, season and municipality, while each experimental variable was measured in triplicate in the laboratory for each real sample.”
- Figure 2: PCA seems confusing, the Principal Component Analysis shows the parameters with plotting of the different honey samples. This cannot provide utile information for the study. The same for figures 3 the correlation analysis is done to show if there is a relation between parameters.
R: The analysis of principal components in this work provides valuable information for the development of the beekeeping industry in Colombia, since it allows establishing the interspecific variability of the measures associated with the quality of honeys produced in humid tropical forests. This, through two interpretations set out below:
- That there are two independent data structures for the honey samples collected by the two species. Which is confirmed with the representation of the variability of both honeys, which is in both cases above 78%, information included in lines 420 to 423. To reach this conclusion, the result of the cluster analysis was considered, in addition to performing a joint PCA with both data structures, which yielded a representation of the variability of less than 50%.
- The magnitude of the vectors of each variable collected in the PCA indicates its contribution to the variability of the system, therefore its recognition allows us to observe that there are quality variables with high interspecific variability for the honeys produced by the different species of bees, regardless from its place of production. This is essential, since in Colombia there is no specific regulation for honeys that are not produced by Apis melifera even though their characteristics are notably different. This information is included in the document between lines 424 to 463.
“Figure 3a shows that in the case of A. mellifera, the main component one is associated with the mineral content of honey, antioxidant capacity, and total polyphenol content. Com-ponent 2 is mainly associated with the ° Brix content and humidity. Additionally, it can be observed that the variables that contribute in greater proportion to the variability both in factors associated with quality and in their biological activities of A. mellifera honey are: D(+)-saccharose, ° Brix, HMF, humidity, and inhibitory capacity of Listeria monocytogenes. These results are like those found by Conti et al [21] where it is shown that in a multi-variate way, humidity and sugar content, accompanied by some minerals such as Magnesium and Potassium are quality variables that contribute a greater proportion to the variability of South American unifloral and multi-floral honey, which is directly related to the characteristics of the polem minerals that contain the local floral species.
Regarding the PCA obtained for the honey collected from M. eburnea (Figure 3b), it is observed that component one is highly related to the DPPH radical trapping activity, pH, and electrical conductivity, these variables being the ones that contribute most to the variability. of the samples. For its part, component two is related to the inhibitory activity of Listeria monocytogenes and the scavenging capacity of ABTS radicals. For this species, the variables that contribute in greater proportion to the variability of quality are related to pH, electrical conductivity, and the capacity to capture radicals DPPH and ABTS ac-companied by the content of total polyphenols. This indicates high interspecific variability in the functional properties of the honey produced by stingless bees, which could be explained by the incipient research on quality improvement regarding the homogeneity, stability, and biochemical properties of the honey produced by stingless bees, accom-panied by its lack of control for its sale and incipient regulations in force worldwide, as reported by Braghini et al [57]. Additionally, some probable correlations are appreciated given the closeness of the vectors for both kinds of honey, a hypothesis that was confirmed by the analysis shown in the following section.”
Likewise, the analysis of correlations allows establishing the possible causal relationships of a variable with respect to another measure for each type of honey, which differ in some of them and the magnitude of their direct and inversely proportional relationship. This information is exposed in detail on lines 465 to 511.
” The analysis of independent correlations between the variables studied in A. Mellifera and M. Eburnea honey are shown in Figures 4a and 4b respectively. In the case of honey produced by A. Mellifera, free acidity had a significant direct relationship with lactonic acidity and ash. It is similar to Mulugeta et al [58] and free acid had a negative correlation with Brix and Sucrose. In M. Eburnea, this variable was directly related to HMF, moisture, lactonic acidity, and ABTS, and negatively related to DPPH, Brix, and pH. The last one was according to other authors. It is due to the organic acid containing such as gluconic acid that increases ion H+ and decreases pH [59].
Regarding color, honey obtained from A. Mellifera had positive relationships with TPC, FRAP, minerals, HMF, free acidity, and lactonic acidity, and negative relationships with sucrose. Honey from M. eburnea had positive relationships with TPC, minerals, glucose, and pH and negative correlations with sucrose, electrical conductivity, and S. typhimurium inhibition. Some authors report a high correlation between color, FRAP, and phenols. It could be explained by the color of honey being influenced by pigments, phenols, minerals, HMF, and products of the Maillard reaction and they present antioxidant ac-tivity [60] [61].
Sucrose concentration for A. Mellifera showed positive relationships with fructose and pH and negative relationships with ABTS, free acidity, TPC, lactonic acidity, color, and Listeria inhibition. For M. eburnea honey, sucrose had only positive correlations with Brix and negative correlations with TPC, ABTS, color and glucose, and free acidity. Electrical conductivity showed positive correlations with fructose and HMF. Negative correlations are presented with DPPH and lactonic acidity. The electrical conductivity of M. eburnea showed positive correlations with free acidity and HMF. Negative correlations were found with DPPH, pH, TPC, ABTS, color, ash, and FRAP. The measurement of electrical conductivity depends on the free acid of the honey. The higher the acid content, the higher the resulting conductivity. This is agreed with reported by Mulugeta et al [58].
In the case of the S. typhimurium inhibition variable for A. Mellifera, it did not show statistically significant relationships with any measured variable, and in the case of M. eburnea honey, it showed negative correlations with color and Brix. The ABTS variable for A. Mellifera honey shows positive correlations with FRAP, TPC, and Listeria inhibition and negative correlations with sucrose, fructose, and pH. Regarding ABTS correlations for M. Eburnea honey, it shows positive correlations with TPC, ash, moisture, lactonic acidity, free acidity, and HMF and negative correlations with electrical conductivity, sucrose, fructose, and L. monocytogenes inhibition.
There was high variability in the results and a high correlation of phenols with FRAP, ABTS, and DPPH which explains why the antioxidant activity was due to the presence of secondary metabolites such as phenolic compounds. These phenolic compounds come from nectar sources, which varied throughout the evaluated seasons, as well as the diversity of plants visited by bees [8]. Results showed that higher levels of these phytochemicals could produce a more potent antibacterial effect against L. monocytogenes. Some authors showed that there is a high correlation between phenolic compounds such as quercetin, rutin, and chlorogenic acid with antibacterial activity, showing greater inhibition in Gram-positive bacteria than in Gram-negative bacteria [3].”
Sincerely, on behalf of all authors,
Isabel Cristina Zapata-Vahos
Associate professor
Faculty of Health Sciences, Universidad Católica de Oriente.
Rionegro- Colombia

Reviewer 2 Report
Abstract
ln.28, analyzed by LDA (Linear Discriminant Analysis)...
Materials and methods
2.2. Obtaining the samples.
Complete the Table 1 with a drawing/Scheme/figure and information about the different characteristics of the A. mellifera and the M. eburnea, to help readers understand why the type of honey will be also different according to they type of bee.
2.3.1. 5-hydroxymethylfurfural (HMF), indicate in subtitle, it is a classical analysis, but for non-familiar readers, it would help, specially if explaining better the connection with the sugars in the samples.
2.4.4. Total phenolic content, as reducing capacity tests
2.5. Antimicrobial activity
2.6. Statistical analysis
Table 2. Please arrange the column of treatments for better appearance.
The meaning of the lowercase letters or the uppercase letters in the Table 2 is not clear, is not presented in the legend.
Figure 3 is of unacceptable quality (difficult to read the numbers and in black and white paper, impossible, please, choose a different combination of colours that would facilitate reading in any conditions (either colour or b/w).
Author Response
Response to reviewers
Manuscript ID: foods-2120784
Manuscript title: Physicochemical parameters, antioxidant capacity and antimicrobial activity of honeys from tropical forests of Colombia: Apis mellifera and Melipona eburnean
Dear Prof. Dr. Arun K. Bhunia
Editor-in-Chief
Foods
On behalf of all the co-authors of this document, I would like to thank the work and time of the jurors in reviewing in detail and contributing to the improvement of the quality and clarity of our work. We find the reviews helpful and have responded to each of your comments in detail. Based on the reviewer's comments, we made modifications to the original manuscript and carefully revised it.
We detail our responses to each of the comments below, and we believe that the manuscript has been greatly improved by your input. We hope that this contribution will be to the liking and approval of the reviewers for publication in their journal.
For the answers and modifications, we refer in each of them to the line numbers. In addition, the language of the document was thoroughly reviewed and corrected as requested, a process carried out by an English language expert.
Response to reviewer 2 comments:
- Line 28, analyzed by LDA (Linear Discriminant Analysis).
R: Reviewer's suggestion accepted and corrected in document text.
- Complete Table 1 with a drawing/Scheme/figure and information about the different characteristics of the mellifera and the M. eburnea, to help readers understand why the type of honey will be also different according to the type of bee.
R: The reviewer's suggestion is accepted, and Table 1 is replaced by an illustrative diagram that shows both the sampling locations and the morphological differences of the two bee species and is complemented with a botanical description of the diversity of pollen found in honey from A. mellifera and M. eburnea from tropical forests (previously published article, derived from the same research).
- 3.1. 5-hydroxymethylfurfural (HMF), indicate in subtitle, it is a classical analysis, but for non-familiar readers, it would help, especially if explaining better the connection with the sugars in the samples.
2.4.4. Total phenolic content, as reducing capacity tests
2.5. Antimicrobial activity
2.6. Statistical analysis
R: The reviewer's suggestion is accepted, the quantification of 5-hydroxymethylfurfural (HMF) is included as a subtitle of the methodology (lines 116 to 124), and a description of the synthesis of HMF and its chemical formation bases is included (line 299 to 303). Additionally, the following subtitles are renumbered and renamed.
- Table 2. Please arrange the column of treatments for better appearance.
R: The reviewer's suggestion is accepted, Table 2 is redesigned to make the difference between the treatments clearer, likewise the names of the variables are resized to identify their values.
- The meaning of the lowercase letters or the uppercase letters in the Table 2 is not clear, is not presented in the legend.
R: The authors express their apologies, due to editing format errors, some of the letters in Table 2 remained as superscripts. This error has already been corrected. In this case, different letters indicate significant differences according to the KW test.
Sincerely, on behalf of all authors,
Isabel Cristina Zapata-Vahos
Associate professor
Faculty of Health Sciences, Universidad Católica de Oriente.

Round 2
Reviewer 1 Report
After careful evaluation of the revised paper, the authors responded to all requested corrections. I have no further comments for the authors regarding the current version. The manuscript can be considered for publication.